# Assessment and Evaluation of Psychological Status of Undergraduate College Students during COVID-19 Pandemic: A Cross-Sectional Study in the United Arab Emirates

**DOI:** 10.3390/ijerph191912487

**Published:** 2022-09-30

**Authors:** Heyam F. Dalky, Yousef M. Aljawarneh, Lubna M. Rajab, Salma Almas, Feddah Al Mazemi, Latifa Al Ali, Sana Abdulghani, Shamma Al Shamsi

**Affiliations:** 1Community and Mental Health Department, Faculty of Nursing, Jordan University of Science & Technology, P.O. Box 3030, Irbid 22110, Jordan; 2Health Sciences Division, Nursing Program, Higher Colleges of Technology, P.O. Box 1626, Fujairah 15825, United Arab Emirates

**Keywords:** anxiety, COVID-19, depression, stress, United Arab Emirates, university students

## Abstract

The COVID-19 pandemic caused by the novel coronavirus instigated a worldwide lockdown that affected students mitigating various psychological issues including depression, stress, and anxiety. This study aimed to assess the impact of COVID-19 pandemic on undergraduate university students’ psychological status in terms of depression, anxiety, and stress. A total of 206 students from the Higher College of Technology (HCT), Sharjah Campuses participated in this descriptive cross-sectional study. Data were collected between March and May 2021. Participants completed an online survey including a demographic data questionnaire and the depression, anxiety, stress scale (DASS-21). The mean depression, stress, and anxiety scores were 15.56 (±11.573), 17.13 (±10.946), and 14.90 (±10.523) respectively. Categorically, most students (33.3%) reported no depression, while 26.1% of students reported moderate depression. For stress, the majority (44.4%) experienced no stress, while 19.8% reported moderate stress. Strikingly, 36.7% of students reported extreme severe anxiety, with 28% reporting no anxiety. Students with history of depression, stress, and anxiety symptoms reported a statistically significant mean difference in depression, stress, and anxiety compared with those with no previous history of those symptoms. We conclude with a recommendation to expand mental health screening among undergraduate university students and design appropriate therapeutic modalities.

## 1. Introduction

The coronavirus disease 2019 (COVID-19) is a highly contagious infectious virus that was first identified in late December 2019 from Wuhan, China and then spread globally to over 200 countries. The World Health Organization (WHO) announced the situation as a global pandemic on 11 March 2020. As of June 2022, over 529 million confirmed cases and over six million COVID-19 associated deaths have been documented worldwide [1].

In the wake of the COVID-19 outbreak, a world-wide lockdown was initiated in consort with massive closures of major places such as schools, universities, malls, and restaurants, causing individuals to adjust their social occupational and familial lifestyle [2]. This created a sense of uncertainty and unpredictability, thereby taking a toll on people’s psychological status. Prominently, the negative influence of the COVID-19 pandemic on mental health has become a matter of concern for clinicians and researchers. Research studies were dedicated to explore the impacts of psychological distress during the spread of COVID-19 infection and lockdown at the individual, community, and global levels [3]. Evidently, the prevalence rates of psychological issues in the general population were found to be higher during COVID-19 pandemic. Globally, in the combined study population of 113,285 individuals, the prevalence of depression, anxiety, and stress was 20%, 35%, and 53%, respectively [4]. In another global online survey, 77% of the participants had stress, 35% of respondents had depression, and 59% of the participants had anxiety [5]. In a systematic review and meta-analysis, the prevalence of stress in five studies with a total sample size of 9074 was 29.6%, the prevalence of anxiety in 17 studies with a sample size of 63,439 was 31.9%, and the prevalence of depression in 14 studies with a sample size of 44,531 people was 33.7% [6]. Among Arab Youth, the psychological impact of the pandemic was manifested by depression, anxiety, and stress with estimated prevalence of 57%, 40.5%, and 38.1%, respectively [7].

Experientially, students and teachers have suffered more psychological distress, such as anxiety, depression, and stress, and students exhibited need for psychological help [3]. Particularly, university students have faced intensified stress due to campus evacuations and the transition to remote learning [8]. The educational process and learning style for students was totally changed requiring extensive exhaustion of resources increasing their vulnerability [9]. The experience of semester disruption was coupled with negative emotions including, but not limited to, isolation and frustration. Moreover, undergraduate university students were distressed over their academic progress in light of changes on teaching and assessment modes. These intense feelings are further fueled by the threat of the virus affecting livelihood and wellbeing of loved ones, resulting in various psychological issues such as depression, anxiety, and stress [3]. Among international higher education students, a systematic review found that the prevalence of depression and anxiety were 34% and 32%, respectively [10], whereas 51% of Emirati university students were found to be in psychological distress [11]. Experts have highlighted the dire need for mental services and have urged for immediate action to monitor and control the pandemic’s negative effects among undergraduate university students [12]. Therefore, the current study aimed at investigating the impact of COVID-19 on the psychological status of undergraduate university students in the United Arab Emirates.

## 2. Materials and Methods

This study employed a cross-sectional design to assess psychological status among undergraduate university students at Higher College of Technology (HCT), Sharjah campuses. A total of 3106 students from all programs received online survey containing the study measuring questionnaires. The included participants were to be above 18 years old and currently studying at Sharjah campuses. Consent was obtained from each participant prior to enrollment in the study. The study was approved by the HCT Research and Ethical Integrity Committee (REIC).

The purpose of the study was to identify the impact of COVID-19 pandemic on HCT students’ psychological status in the form of depression, anxiety, and stress. A demographic data questionnaire with 12 questions was developed by the authors of this study collect participants’ demographic variables, history of depression, anxiety, and stress symptoms, and history of psychiatric therapy. The Depression, Anxiety and Stress Scale-21 Items (DASS-21) was used to measure the psychological status of the students in form of depression, anxiety, and stress. The DASS is a self-report scale designed to measure the negative emotional status. The original scale consists of 42 items; however, for the purpose of this study, the short form (21-items) was utilized [13]. The DASS-21 is a short-form scale containing 21 items with three subscales as follows: depression subscale (7 items), anxiety subscale (7 items) and stress subscale (7 items). Each of the three DASS-21 scales contains 7 items, divided into subscales with similar content. The depression scale assesses dysphoria, hopelessness, devaluation of life, self-deprecation, lack of interest/involvement, anhedonia, and inertia. The anxiety scale assesses autonomic arousal, skeletal muscle effects, situational anxiety, and subjective experience of anxious affect. The stress scale is sensitive to levels of chronic nonspecific arousal. It assesses difficulty relaxing, nervous arousal, and being easily upset/agitated, irritable/over-reactive, and impatient. Scores for depression, anxiety, and stress are calculated by summing the scores for the relevant items and then multiplying by two [13]. Respondents rate the presence of each symptom during the past week on a scale from 0 to 3. Correspondingly, 0 denotes: Did not apply to me at all; 1 denotes: Applied to me to some degree, or some of the time; 2 denotes: Applied to me to a considerable degree or a good part of time; and 3 denotes: Applied to me very much or most of the time. Scores for each subscale classified into 5 levels: normal, mild, moderate, severe, and extremely severe. For Depression subscale, normal: 0–9; mild: 10–13; moderate: 14–20; severe: 21–27; and extremely severe: 28 and above. For Anxiety subscale, normal: 0–7; mild: 8–9; moderate: 10–14; severe: 15–19; and extremely severe: 20 and above. For Stress subscale, normal: 0–14; mild: 15–18; moderate: 19–25; severe: 26–33; and extremely severe: 34 and above [13].

Evidently, DASS-21 has proven psychometric properties with a satisfactory discriminant reliability of Cronbach’s alpha values. Cronbach’s alpha was 0.942 for the whole scale and was 0.874, 0.876, and 0.875 for depression, anxiety, and stress subscales [14].

Due to the transition to online learning and teaching imposed during COVID-19 pandemic, the demographic data questionnaire and the DASS-21 were converted into an online survey using Google Docs and sent to the potential participants using their university email address. The survey link includes a digital consent form and the participants’ information sheet. Participation was completely voluntary, and participants had the right to withdraw from the study at any time and without reason. Participation was anonymous, and all responses were kept confidential in a secured and password protected device. There was no anticipated physical or psychological harm from participating in this study with no academic ramification from not participating in the study. Data were collected between March and May 2021. A 1-month email reminder was sent to all potential participants after the initial dissemination of the study survey. All completed surveys were received by the Principal Investigator (PI) and screened for incomplete or missing data.

All data were statistically analyzed using the SPSS software (IBM SPSS Statistics, Version 28, 2021 (SPSS Inc., Chicago, IL, USA). All statistical analyses were performed on the 207 students who met the study inclusion criteria. All data were examined for violation of statistical assumptions prior to each analysis. Descriptive statistics were computed in terms of frequency and percentages for categorical variables, and means and standard deviations (SD) for continuous variables. The chi-squared test was used to compare the differences in some demographic variables with the students’ depression, anxiety, and stress categories. Multiple independent sample Student’s *t*-tests were performed to compare the means of the depression scores, the anxiety scores, and the stress scores between males and females, between students with history of anxiety and those with no history of anxiety, between students with history of depression and those with no history of depression, and between students who visited a psychiatrist in last six months and those who did not. For all statistical analyses, a *p*-value of less than 0.05 was used as a cut-off for statistical significance level.

## 3. Results

### 3.1. Participants Characteristics

Of all sent invites to participate in the study, 207 students from HCT Sharjah Women’s Campus participated in the study with a modest response rate (62%). The mean age of the participants was 20.87 (±2.68) years. The sample was primarily female (n = 196, 94.7%), single (n = 181, 87.4%), unemployed (n = 190, 91.8%), from health sciences division (n = 91, 44.0%), and in their 4th year of study (n = 72, 34.8%). More than half of the participants reported a Grade Point Average (GPA) of 2.0–2.9 (n = 113, 54.6%), and 81 of them (38.1.0%) reported a good academic standing with a GPA of more than 3.0. In contrast, the majority of the participants reported studying less than nine hours per week (n = 102, 49.3%).

The demographic questionnaire included four psychological-related questions. The first question asked the participants if they experienced anxiety symptoms in the past six months. Most of the participants (n = 126, 60.9%) reported having anxiety symptoms in the past, while 22 of them (10.6%) preferred not to tell. The second question asked the participants if they experienced depression symptoms in the past six months. The responses were almost evenly distributed between having depression symptoms in the past or not (n = 95, 45.9%; n = 93, 44.9% respectively), while 17 of them (8.2%) preferred not to tell. In the third question, the participants were asked if they visited a family physician in the last two weeks for the experienced anxiety and depression symptoms, and the majority of the participants answered with no (n = 155, 74.9%), while 17 of them preferred not to tell (8.2%). The last question asked the participants if they visited a psychiatrist in the last six months. The majority of the participants have not visited a psychiatrist in the last six months (n = 173, 83.6%), while 13 of them preferred not to tell (6.3%). The demographic characteristics and the psychological-related questions are presented in Table 1.

### 3.2. Descriptive Statistics of the DASS

Participants’ depression, stress, and anxiety levels were measured using the DASS-21. The data were firstly analyzed following the continuous scoring protocol and reported in mean ± standard deviation (SD). The DASS data among the participants showed a mean depression score of 15.56 (±11.573), a mean stress score of 17.13 (±10.946) and a mean anxiety score of 14.90 (±10.523) as shown in Table 2.

Second, the DASS data were converted into categories for each of the subscales following the categorical scoring protocol and the data is shown in Table 3. For depression, 39 students (18.8.1%) had extremely severe depression, 30 students (14.5%) had severe depression, 54 students (26.1%) had moderate depression, 14 students (6.8%) had mild depression, and 70 students had no depression (33.8%). Regarding stress, the majority of the students had no stress (n = 92, 44.85%), while 16 students (7.7%) had extremely severe stress, 36 (17.4%) had severe stress, 41 (19.8%) had moderate stress, and 22 students (10.6%) had mild stress. For anxiety, 36.7% of the students had extremely severe anxiety, 10.6% had severe anxiety, 17.9% had moderate anxiety, and 6.8% had mild anxiety. On the other hand, 28% of the students had no anxiety.

Multiple Pearson’s chi-squared tests were computed. Due to significantly imbalanced sample distribution between females and males (196:11), the Pearson’s chi-squared test was not computed to assess the association between gender and the participants’ categories of depression, stress, and anxiety. The first Pearson’s chi-squared test was computed to assess the association between participants’ depression category and stress category. The statistical assumptions for Pearson’s chi-squared test were examined prior each analysis. A statistically significant association between depression category and stress category was observed: χ^2^(16) = 185.19, *p* < 0.001 (Table 4, Figure 1). This association between the depression category and the stress category was strong and positive: r (16) = 0.808, *p* < 0.001.

Another Pearson’s chi-squared test was computed to assess the association between participants’ depression category and anxiety category. The results revealed a statistically significant association between the depression category and the anxiety category: χ^2^(16) = 198.58, *p* < 0.001. The association between the two variables was strong and positive: r(16) = 0.802, *p* < 0.001 (Table 5, Figure 2).

The Pearson’s chi-squared test on the association between the stress category and the anxiety category showed a statistically significant association: χ^2^(16) = 135.62, *p* < 0.001. The association between the two variables was strong and positive: r(16) = 0.719, *p* < 0.001 (Table 6, Figure 3).

The first independent samples Student’s *t*-test to compare the means of the depression scores, the stress scores, and the anxiety scores between males and females showed no statistically significant mean difference in depression scores between female (15.35.± 11.638) and male students (19.27 ± 10.090) with t(205) = −1.095, *p* = 0.275; no statistically significant mean difference in stress scores between female (16.97 ± 10.996) and male students (20.00 ± 10.040) with t(205) = −0.893, *p* = 0.373; and no statistically significant mean difference in anxiety scores between female (14.66 ± 10.553) and male students (19.09 ± 9.439) with t(205) = −1.361, *p* = 0.175 (Table 7).

The independent samples Student’s *t*-test to compare the means of the depression scores, the stress scores, and the anxiety scores between students with history of anxiety symptoms and those with no history of anxiety symptoms showed a statistically significant mean difference in depression scores between students with history of anxiety symptoms (17.83 ± 11.774) and those with no history of anxiety symptoms (12.03 ± 10.619) with t(183) = 3.221, *p* = 0.002; a statistically significant mean difference in stress scores between students with history of anxiety symptoms (19.35 ± 10.582) and those with no history of anxiety symptoms (13.83 ± 10.842) with t(183) = 3.280, *p* = 0.001; and a statistically significant mean difference in anxiety scores between students with history of anxiety symptoms (16.87 ± 10.234) and those with no history of anxiety symptoms (10.95 ± 9.975) with t(183) = 3.699, *p* < 0.001 (Table 8).

The independent samples Student’s *t*-test to compare the means of the depression scores, the stress scores, and the anxiety scores between students with history of depression symptoms and those with no history of depression symptoms showed a statistically significant mean difference in depression scores between students with history of depression symptoms (20.86 ± 10.980) and those with no history of depression symptoms (10.73 ± 10.124) with t(186) = 6.574, *p* < 0.001; a statistically significant mean difference in stress scores between students with history of depression symptoms (21.68 ± 10.107) and those with no history of depression symptoms (12.92 ± 10.226) with t(186) = 5.907, *p* < 0.001; and a statistically significant mean difference in anxiety scores between students with history of depression symptoms (19.16 ± 10.209) and those with no history of depression symptoms (10.80 ± 9.046) with t(186) = 5.947, *p* < 0.001 (Table 9).

The last independent samples Student’s *t*-test to compare the means of the depression scores, the stress scores, and the anxiety scores between students who visited a psychiatrist in last six months and those who did not showed no statistically significant mean difference in depression scores between students who visited a psychiatrists in last six months (20.00 ± 12.977) and those with did not (15.34 ± 11.254) with t(191) = 1.762, *p* = 0.080; no statistically significant mean difference in stress scores between students who visited a psychiatrist in last six months (20.19 ± 11.847) and those with did not (17.10 ± 10.757) with t(191) = 1.227, *p* = 0.221; and a statistically significant mean difference in anxiety scores between students who visited a psychiatrist in last six months (19.90 ± 12.025) and those with did not (14.42 ± 10.074) with t(191) = 2.305, *p* = 0.011 (Table 10).

## 4. Discussion

This research study was conducted in the United Arab Emirates at the Higher College of Technology to assess the impact of COVID-19 pandemic on undergraduate college students’ psychological status. The study found that anxiety was the most prevalent psychological issue (72%). While it has been reportedly consistent in the literature that anxiety is the dominant psychological issue among undergraduate university students, the prevalence in this study was relatively higher than previously recorded data in the United Arab Emirates. The prevalence of anxiety among Emirati undergraduate university students was estimated to be 55% [15]. However, these estimates were from data collected preceding the COVID-19 pandemic confirming the effect of the pandemic on students’ psychological status. During the COVID-19 pandemic, a study on 433 Emirati undergraduate university students showed that only 15.9% had anxiety [11]. However, the study measured anxiety specifically about COVID-19 using the Coronavirus Anxiety Scale. Each item rated on this scale reflects the frequency of anxiety symptoms “when thinking about or being exposed to information about the coronavirus” [11]. In contrast, the same study revealed that more than half of the students experienced psychological distress [11]. While the measure for anxiety was specific to COVID-19, the measure for distress was universal further supporting the explanation for discrepancy. Another United Arab Emirates study assessing anxiety using the generalized anxiety disorder scale showed proportional results to this study with almost half of 1485 sampled Emirati university students experiencing mild to severe anxiety [16]. Similar to DASS-21 the generalized anxiety disorder scale asks participants how often they were bothered by each symptom regardless of a specific stimulant; however, the severity range is limited from mild to severe, not allowing for comparison with the extremely severe rates in this study. A study conducted on 1380 Jordanian undergraduate university students using DASS-21 showed comparable results to our study with 67.9% of the students had anxiety, where 52.8% of them reported anxiety that ranged from mild to severe and 15.1% reported extremely severe anxiety [17].

For depression, the findings were comparable to results in Lebanon showing that 17.9% of 520 Lebanese University undergraduate students had mild depressive symptoms, while 13.8% had moderate and 1.7% had severe depressive symptoms [18]. However, the results from this study were insignificant compared to Saudi Arabia, where 48.8% of undergraduates showed significant symptoms of depression [19]. This may be attributed to the peak in COVID-19 cases at the time, as the study was conducted in earlier stages of the pandemic. It may be thus hypothesized that the COVID-19 pandemic had an effect on depression in the initial outbreak and spread of the virus. This attribution is further reinforced by findings from Asia in the early COVID-19 stages, where 82.4% of students were found to have mild to severe depressive symptoms [20]. Moreover, findings from Jordan showed that the majority of 6157 undergraduates (37.2%) experienced high depression, while 34.1% experienced moderate depression and only 28.7% had no depression [21]. These findings were collected during the extensive quarantine period which may explain the intensity of depressive symptoms pertaining to increased withdrawal and isolation. Similarly, in the United States of America, 48.14% of 2031 undergraduates showed a moderate-to-severe level of depression [22]. This data was collected during the initial peak of the COVID-19 pandemic in the late Spring 2020 semester at Texas after the State of Texas issued a stay-at-home order on April 2, 2020 [22].

Regarding stress, the results were proportional with studies conducted in Lebanon, Egypt, and USA. In Lebanon, the majority (87.3%) of 520 Lebanese University undergraduate students had no stress, while 11% of had mild stress and 1.7% had moderate stress [18]. Similarly, in a sample of 1335 Egyptian undergraduate university students, 52.2% of students had normal stress, 33.6% of students reported mild to moderate stress levels, and 14.2% reported severe to extremely severe stress levels [23]. In the USA, among 676 American undergraduate university students, 67.4% had normal stress, 21.2% of students reported mild to moderate stress levels, and 11.4% reported severe to extremely severe stress levels [24]. However, variation was noted from previously reported stress levels in the UAE where moderate stress was found among 40.7% of the student population [25]. This may be attributed to the sample characteristics, as the sample included a small size of 81 senior pharmacy students. Pharmacy students are especially burdened by healthcare involvement during the pandemic and during their last year students they deal with an additional stressor of job market preparation. Further studies targeted students in the healthcare sector as a vulnerable population categorizing their field of study as a stressor COVID times. In fact, 70.9% of healthcare college students were found to have poor mental health during the pandemic in Japan. [26]. In a study conducted in Saudi Arabia, 38% of 933 health-care undergraduates experienced moderate to extremely severe stress [27]. In Jordan, the mean stress scores 485 healthcare students were at 17.1 (±11.8) using DASS-21 [28]. In fact, nursing students were found to be more stressed during COVID-19 than their counterparts from other majors [29].

This study is not without limitations; while the cross-sectional design serves the purpose of this study, the type of design hinders determining the cause-and-effect relationships between the variables. In addition, the sample size does not allow for generalizability of results. Moreover, since the research involved the participants’ self-report, the risk of information and record bias should be considered. Finally, the imbalance in gender distribution across this sample did not allow for gender comparison among variables.

## 5. Conclusions

In conclusion, COVID-19 pandemic has affected the psychological status of undergraduate university students, mainly manifesting through moderate depression and severe anxiety among the majority of students. A previous history of depression, anxiety, or stress symptoms had an impact on the psychological status of the studied population. This entails considerations for social workers or academic/campus counselors to tackle mental health of their students more extensively. This can be achieved through different campaigns, discussion forums or interactive sessions that promote mental health literacy. Future implications suggest expanding mental health screening across campuses and designing appropriate intervention strategies.

## Figures and Tables

**Figure 1 ijerph-19-12487-f001:**
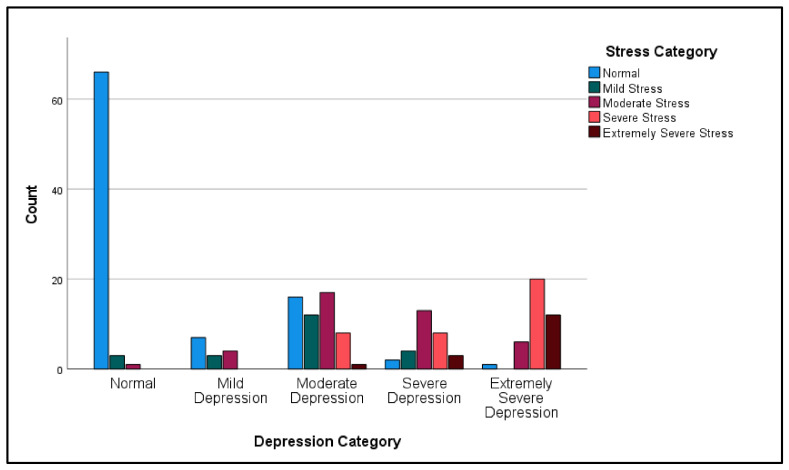
Participants depression category by stress category.

**Figure 2 ijerph-19-12487-f002:**
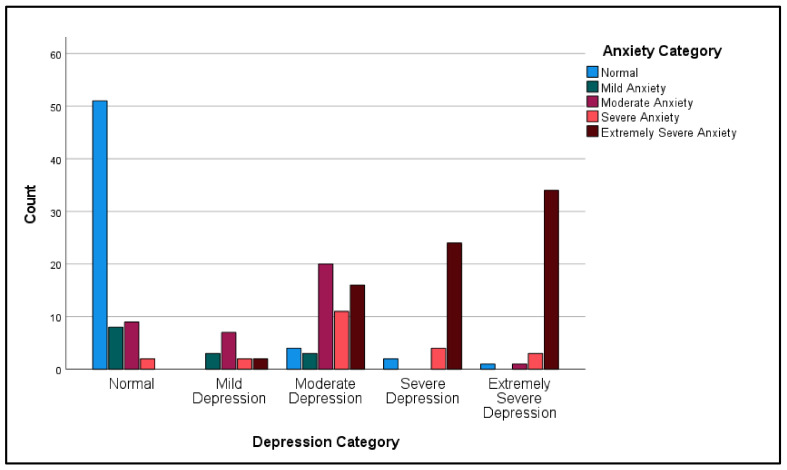
Participants depression category by anxiety category.

**Figure 3 ijerph-19-12487-f003:**
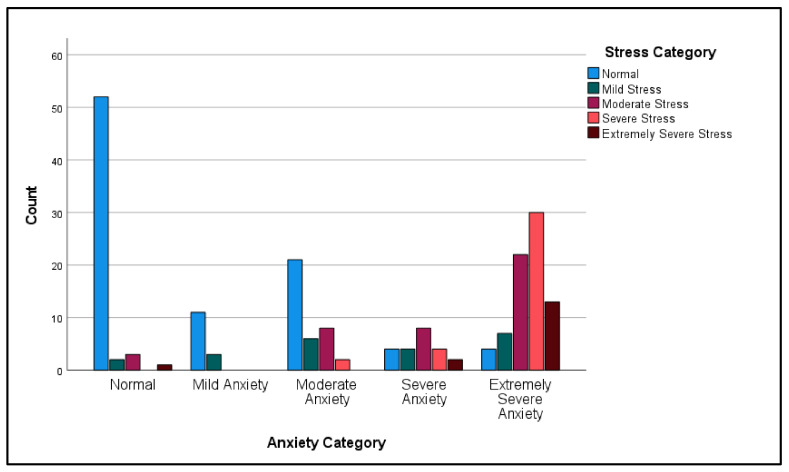
Participants anxiety category by stress category.

**Table 1 ijerph-19-12487-t001:** Demographic Characteristics and Psychological-Related Questions (N = 207).

Variables	Frequency (%)
Gender	
Male	11 (5.3)
Female	196 (94.7)
Academic Division	
Health Science	91 (44.0)
Engineering Technology and Science	20 (9.7)
Education	20 (9.7)
Computer Information Science	32 (15.5)
Business	37 (19.2)
Applied Media	4 (1.9)
Marital Status	
Single	181 (87.4)
Married	18 (8.7)
Divorced	2 (1.0)
Prefer Not to Answer	4 (1.9)
Employment	
Yes	17 (8.2)
No	190 (91.8)
Grade Point Average	
0–1.9	13 (6.3)
2–2.9	113 (54.6)
3–4	81 (39.1)
Weekly Academic Studying Hours	
Less than 9 h	102 (49.3)
9–12 h	68 (32.9)
13–18 h	21 (10.1)
More than 18 h	15 (7.2)
Year of Study	
1st year	49 (23.7)
2nd year	53 (25.6)
3rd year	33 (15.9)
4th year	72 (34.8)
Had Anxiety Symptoms in the Last Six Months	
Yes	126 (60.9)
No	59 (28.5)
Prefer Not to Tell	22 (10.6)
Had Depression Symptoms in the Last Six Months	
Yes	95 (45.9)
No	93 (44.9)
Prefer Not to Tell	17 (8.2)
Visited a Family Physician in the Last Two Weeks	
Yes	35 (16.9)
No	155 (74.9)
Prefer Not to Answer	17 (8.2)
Visited a Psychiatrist in the Last Six Months	
Yes	21 (10.1)
No	173 (83.6)
Prefer Not to Tell	13 (6.3)
Age (M ± SD)	21 (±2.2)

**Table 2 ijerph-19-12487-t002:** Students’ Descriptive Statistics of the DASS (N = 207).

Variables	Mean (±SD)
Depression Score	15.56 (±11.573)
Stress Score	17.13 (±10.946)
Anxiety Score	14.90 (±10.523)

**Table 3 ijerph-19-12487-t003:** Frequencies of Depression, Stress, and Anxiety among Participants (N = 207).

Variables	Frequency (%)
Depression	
No Depression	70 (33.3)
Mild Depression	14 (6.8)
Moderate Depression	54 (26.1)
Severe Depression	30 (4.5)
Extremely Severe Depression	39 (18.8)
Stress	
No Stress	92 (44.4)
Mild Stress	22 (10.6)
Moderate Stress	41 (19.8)
Severe Stress	36 (17.4)
Extremely Severe Stress	16 (7.7)
Anxiety	
No Anxiety	58 (28.0)
Mild Anxiety	14 (6.8)
Moderate Anxiety	36 (17.9)
Severe Anxiety	22 (10.6)
Extremely Anxiety	76 (36.7)

**Table 4 ijerph-19-12487-t004:** Participants Depression Category by Stress Category.

	Stress Category				
Depression Category	No	Mild	Moderate	Severe	Extremely Severe
No Depression	66	3	1	0	0
Mild	7	3	4	0	0
Moderate	16	12	17	8	1
Severe	2	4	13	8	3
Extremely Severe	1	0	6	20	12

χ^2^(16) = 185.19, *p* < 0.001.

**Table 5 ijerph-19-12487-t005:** Participants Depression Category by Anxiety Category.

	Anxiety Category				
Anxiety Category	Normal	Mild	Moderate	Severe	Extremely Severe
No Anxiety	51	8	9	2	0
Mild	0	3	7	2	14
Moderate	4	3	20	11	54
Severe	2	0	0	4	30
Extremely Severe	1	0	1	3	39

χ^2^(16) = 198.58, *p* < 0.001.

**Table 6 ijerph-19-12487-t006:** Participants Anxiety Category by Stress Category.

	Stress Category				
Stress Category	Normal	Mild	Moderate	Severe	Extremely Severe
No Stress	52	2	3	0	1
Mild	11	3	0	0	0
Moderate	21	6	8	2	0
Severe	4	4	8	4	2
Extremely Severe	4	7	22	30	13

χ^2^(16) = 135.62, *p* < 0.001.

**Table 7 ijerph-19-12487-t007:** Independent Samples *t*-tests for Gender on Depression, Stress and Anxiety Scores (N = 207).

Scores	Gender	N	Mean	SD	∆Mean	t	*p*	95% CI	
								LL	UL
Depression	Female	196	15.35	11.638	−3.926	−1.095	0.275	−10.993	3.141
	Male	11	19.27	10.090	−3.926				
Stress	Female	196	16.97	10.996	−3.031	−0.893	0.373	−9.721	3.660
	Male	11	20.00	10.040	−3.031				
Anxiety	Female	196	14.66	10.553	−4.428	−1.361	0.175	−10.843	1.988
	Male	11	19.09	9.439	−4.428				

**Table 8 ijerph-19-12487-t008:** Independent Samples *t*-tests for History of Anxiety on Depression, Stress, and Anxiety Scores (N = 207).

Scores	History of Anxiety Symptoms	N	Mean	SD	∆Mean	t	*p*	95% CI	
								LL	UL
Depression	Yes	126	17.83	11.744	5.791	3.221	0.002	2.243	9.340
	No	59	12.03	10.619	5.791				
Stress	Yes	126	19.35	10.582	5.519	3.280	0.001	2.199	8.838
	No	59	13.83	10.842	5.519				
Anxiety	Yes	126	16.87	10.234	5.924	3.699	<0.001	2.764	9.084
	No	59	10.95	9.975	5.924				

**Table 9 ijerph-19-12487-t009:** Independent Samples *t*-tests for History of Depression Symptoms on Depression, Stress, and Anxiety Scores (N = 207).

Scores	History of Depression Symptoms	N	Mean	SD	∆Mean	t	*p*	95% CI	
								LL	UL
Depression	Yes	95	20.86	10.980	10.132	6.574	<0.001	7.091	13.173
	No	93	10.73	10.124	10.132				
Stress	Yes	95	21.68	10.107	8.759	5.907	<0.001	5.834	11.685
	No	93	12.92	10.226	8.759				
Anxiety	Yes	95	19.16	10.209	8.362	5.940	<0.001	5.585	11.140
	No	93	10.80	9.046	8.362				

**Table 10 ijerph-19-12487-t010:** Independent Samples *t*-tests for History of Psychiatrist Visit on Depression, Stress, and Anxiety Scores (N = 207).

Scores	History of Psychiatrist Visit	N	Mean	SD	∆Mean	t	*p*	95% CI	
								LL	UL
Depression	Yes	21	20.00	12.977	4.663	1.762	0.080	−0.556	9.882
	No	172	15.34	11.254	4.663				
Stress	Yes	21	20.19	11.847	3.086	1.227	0.221	−1.873	8.045
	No	172	17.10	10.757	3.086				
Anxiety	Yes	21	19.90	12.025	5.486	2.305	0.022	0.792	10.180
	No	172	14.42	10.074	5.486				

## Data Availability

The data presented in this study are available on request from the corresponding author.

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
