# Peer review of "Assessment and Evaluation of Psychological Status of Undergraduate College Students during COVID-19 Pandemic: A Cross-Sectional Study in the United Arab Emirates"

_ijerph, 2022, doi:10.3390/ijerph191912487_

Round 1

Reviewer 1 Report

Dear authors, 

Thank you for allowing me to review this article. I have a few suggestions :

1) The title seems rather confusing and grammatically incorrect. Please revise

2) The term 'normal depression' and 'normal stress' is confusing and should be rephrased as 'no depression' and 'no stress'.

3) In the discussion, the authors should explain in more detail why they found ( rather surprisingly ) that the rates of depression, anxiety and stress were low among their sample population. This is in stark contrast to the literature available. Could this be due to methodological issues ?

5) Generally, the discussion section should consist of critical analysis of the study results. However, the authors seemed to have repeated the result findings into their discussion. The authors are advised to restructure their discussion to make it more impactful to readers.

4) In the introduction, the authors are suggested to include other global studies studying the impact of COVID-19. The authors are guided to this reference:

Narendra Kumar, M.K., Francis, B., Hashim, A.H., Zainal, N.Z., Abdul Rashid, R., Ng, C.G., Danaee, M., Hussain, N. and Sulaiman, A.H., 2022, March. Prevalence of Anxiety and Depression among Psychiatric Healthcare Workers during the COVID-19 Pandemic: A Malaysian Perspective. In Healthcare (Vol. 10, No. 3, p. 532). MDPI. 

Author Response

Dear Reviewer,

Thank you for your feedback and valuable comments. All your comments have been addressed where applicable/needed and the manuscript has been revised accordingly.

  1. The title seems rather confusing and grammatically incorrect. Please revise. Revised accordingly.
  2. The term 'normal depression' and 'normal stress' is confusing and should be rephrased as 'no depression' and 'no stress'. Changed accordingly.

  3. In the discussion, the authors should explain in more detail why they found ( rather surprisingly ) that the rates of depression, anxiety and stress were low among their sample population. This is in stark contrast to the literature available. Could this be due to methodological issues ? Revised accordingly.

  4. Generally, the discussion section should consist of critical analysis of the study results. However, the authors seemed to have repeated the result findings into their discussion. The authors are advised to restructure their discussion to make it more impactful to readers. Restructured where applicable.

  5. In the introduction, the authors are suggested to include other global studies studying the impact of COVID-19. The authors are guided to this reference: The impact of COVID-19 pandemic on the psychological status at academic venues was enriched from global perspectives.

Please find all the changes and revisions highlighted in the attached revised copy of the manuscript.

Regards

Reviewer 2 Report

It is a pleasure to review this manuscript entitled

 Psychological Assessment “Depression, Stress and Anxiety” of 2 Undergraduate Colleges Students During COVID-19 Pandemic: A Cross-sectional Study in the UAE, an interesting topic for general and future students.

After reviewing the document, I propose small changes to enrich your manuscript.

In abstract and line 298 you refer and consider stress, anxiety and depression as a "normal" entity, there is no "normal" category. Please modify it to the corresponding degree.

Line 48- Among the general population, it was experiential that students and teachers suffered more psychological distress, such as anxiety, depression, and stress and students exhibited needs for immediate psychological help [3].

It is necessary to point out that students and teachers are a group that should not be included in the general population; it is convenient to modify the sentence for the clear understanding of the reader. In addition, it would be convenient to reword the sentence that students exhibited needs for immediate psychological help according to Cameron's article, which talks about the need for changes but not for immediate psychological help.

Line 58- the quote from cameron et al, 2020, reference it according to journal guidelines.

Line 65 correct COVD-19

Thank you, I hope you can answer and modify them.

Author Response

Thank you for your feedback and valuable comments. All your comments have been addressed where applicable/needed and the manuscript has been revised accordingly.

  1. In abstract and line 298 you refer and consider stress, anxiety and depression as a "normal" entity, there is no "normal" category. Please modify it to the corresponding degree. Modified accordingly.
  2. Line 48- Among the general population, it was experiential that students and teachers suffered more psychological distress, such as anxiety, depression, and stress and students exhibited needs for immediate psychological help [3]. Revised accordingly.

  3. It is necessary to point out that students and teachers are a group that should not be included in the general population; it is convenient to modify the sentence for the clear understanding of the reader. In addition, it would be convenient to reword the sentence that students exhibited needs for immediate psychological help according to Cameron's article, which talks about the need for changes but not for immediate psychological help. Revised accordingly.

  4. Line 58- the quote from cameron et al, 2020, reference it according to journal guidelines. Revised accordingly.

  5. Line 65 correct COVD-19. Corrected accordingly.

Please find all the changes and revisions highlighted in the attached revised copy of the manuscript.

Regards

Round 2

Reviewer 1 Report

Thanks for the revisions. No further comments